

# Temporal trends of land-use favourability for the strongly declining little bustard: assessing the role of protected areas

David González del Portillo[1], Manuel B. Morales[1] and Beatriz Arroyo[2]

[1] Department of Ecology and Research Center on Biodiversity and Global Change, Autónoma University of Madrid, Madrid, Spain
[2] Instituto de Investigación en Recursos Cinegéticos (IREC, CSIC-UCLM-JCCM), Ciudad Real, Spain

Corresponding author
David González del Portillo,
david.gonzalezd01@estudiante.uam.es,
davigo08@ucm.es

## ABSTRACT

The little bustard (*Tetrax tetrax*) is a steppe bird strongly and negatively influenced by agricultural intensification in Europe. Here, we use the little bustard as a model species to examine how favourability (relative occurrence likelihood of a species based on environmental characteristics, such as habitat availability) varies regionally with degree of protection in north-western Spain. The Natura2000 network is one of the main biodiversity conservation tools of the European Union, aiming to protect areas hosting species of conservation concern from unfavourable land-use changes. The network covers many landscapes across the continent, including farmland. Additionally, we examine the relationship between trends in land-use favourability and little bustard population trends over a decade in the Nature Reserve of Lagunas de Villafáfila, a protected area also in the Natura2000 network where active and intense management focused on steppe bird conservation is carried out. Favourability was much greater in Villafáfila than in both protected areas with lower degree of protection and in non-protected areas. Land-use favourability increased slightly between 2011 and 2020 both in and out of protected areas, whereas little bustard populations declined sharply in that period, even in Villafáfila. Spatial variations in little bustard abundance within Villafáfila depended on social attraction (increasing with the number of neighbouring males) but not significantly on small-scale variations in land-use favourability. These results suggest that land-use management in Natura2000 areas needs to be more conservation-focused, favouring natural and seminatural habitats and traditional farming practices to improve land-use favourability for little bustards and other steppe birds. Additional factors, such as field-level agricultural management or social interaction variables that may cause an Allee effect, should be incorporated in little bustard favourability models to improve their use in conservation planning.

## INTRODUCTION

Land-use change is among the main causes of biodiversity loss (*Díaz et al., 2019*) due to an associated decrease in habitat suitability for many species (*Thuiller, 2007*). In many places today, land use in agriculture is becoming more intensive to increase yield and income.

This intensification tends to be associated habitat loss for farmland wildlife. Farmlands may also, but less often, be abandoned and that may also bring about a loss of habitat quality for farmland wildlife through shrub encroachment (*Suárez-Seoane, Osborne & Baudry, 2002*; *Emmerson et al., 2016*).

Populations of threatened steppe bird species are often found in European cereal farmland, particularly in the Iberian Peninsula (*Santos & Suárez, 2005*), and are thus vulnerable to the effects of land-use changes occurring there. The establishment and maintenance of protected areas is among the most important conservation policy tools used to tackle land-use change impacts on biodiversity around the globe. The Natura2000 network, for example, is the main land-planning tool for biodiversity protection in the European Union, covering many landscapes across the continent (including farmland) with the aim to protect important biodiversity areas and make their conservation compatible with existing land-use (*European Commission, 2022*). However, the establishment and management of Natura2000 sites depends on member states (which, as in Spain, may delegate these competences to regional administrations). When each country or region independently administrates its protected sites, differences among them may lead to variation in effective implementation and jeopardize the common goal of maintaining and restoring European habitats or species (*McKenna et al., 2014*).

Spain's Natura2000 network (which covers around 20% of the country) includes farmland with important populations of threatened steppe bird species included in Annex I of the Birds Directive (*Ministry for Ecological Transition MITECO, 2021*). However, Natura2000 sites such as Special Protection Areas (hereafter SPAs) have been shown to be inefficient in protecting farmland biodiversity against land use changes and agriculture intensification (*Palacín & Alonso, 2018*; *Gameiro et al., 2020*). On the other hand, Natura2000 areas coexist (and often overlap) with member-state-level protection sites, such as nature reserves. Nature reserves in Spain have been suggested to be more effective against land use and cover changes than SPAs due to their more stringent and conservation-oriented management, compared to the less effective measures of the Natura2000 network (*Rodríguez-Rodríguez & Martínez-Vega, 2018*). Further, SPAs frequently lack regular or systematic biodiversity monitoring and assessment of management results, which are paramount to develop evidence-based conservation programs (*Kleijn & Sutherland, 2003*; *Trochet & Schmeller, 2013*).

Various approaches are used to measure the effectiveness of protected areas, such as directly measuring changes in species' abundance or in land use cover. An alternative is to measure changes in habitat quality or suitability for a given species or group of species. The latter is particularly relevant when species may use a variety of land-use types, so that changes in one do not necessarily lead to an overall loss or gain in habitat quality. Habitat suitability may be calculated from occurrence modelling to determine the variables that increase the probability of presence of a given species (*Guisan & Thuiller, 2005*; *Miller, 2010*). In this context, the habitat favourability function is of particular interest and has been widely used in species distribution modelling, habitat selection and epidemiology (*Manly et al., 2007*; *Pfeiffer et al., 2008*; *Franklin, 2010*). The main advantage of the favourability function is that it allows direct comparison between different samples (years, areas, or

species) regardless of species prevalence (*Acevedo & Real, 2012*). In addition, favourability indices may be extrapolated in space and time independently of variations in prevalence, thus allowing for the assessment of changes over time or across areas with a single indicator (*Acevedo & Real, 2012*).

The little bustard (*Tetrax tetrax*) is a farmland steppe bird strongly and negatively affected by recent agricultural changes in Europe (*Traba & Morales, 2019*; *Morales & Bretagnolle, 2022*; *Santangeli et al., 2023*). This bustard occupies extensive, heterogenous farmland landscapes (*Morales, García & Arroyo, 2005*; *Faria & Silva, 2010*) and can be an indicator of well conserved agricultural steppe ecosystems and an umbrella species for other steppe birds (*Morales & Bretagnolle, 2022*; *Morales, Merencio & García de la Morena, 2023*). The little bustard is in strong decline all over Europe (*e.g.*, *Morales & Bretagnolle, 2022*; *Santangeli et al., 2023*) associated with increasing agricultural intensification (*Inchausti & Bretagnolle, 2005*; *Traba & Morales, 2019*) which typically includes reduced fallow surface, increased irrigation, and monocultures leading to landscape simplification, as well as an increase in chemical inputs (*Matson et al., 1997*; *Emmerson et al., 2016*; *Stanton, Morrissey & Clark, 2018*), all of which reduce habitat suitability. Some farming practices (such as fallow ploughing or night operations) are particularly detrimental for the little bustard, causing nest loss, nestling and adult mortality (*Morales et al., 2013*; *Bretagnolle, Denonfoux & Villers, 2018*; *Silva et al., 2022*), while the use of agrochemicals (*i.e.,* fertilizers or pesticides) may have effects on food abundance.

Here, we (a) examine land-use variables that determine habitat favourability for the little bustard in the extensive cereal farmland of the Duero basin (NW Spain), and (b) compare favourability and its temporal trends between areas of different protection levels (unprotected, SPAs, a nature reserve), to determine the effectiveness of the Natura2000 program. If Natura2000 policies are effective, then favourability will be associated with degree of protection, and will be higher and most stable in most protected areas. Additionally, we (c) compare little bustard relative abundance among areas with different level of protection to test whether little bustard abundance is related to favourability and protection level. Finally, we (d) use 10 years of breeding censuses data to test whether little bustard population trends match those of habitat favourability in a protected area. If habitat quality loss is a driver of the little bustard decline, the latter should be associated with declining habitat favourability.

## MATERIALS AND METHODS

### Study species

The little bustard is a medium-sized sexually dimorphic steppe bird that inhabits natural grasslands and farmlands (*Cramp & Simmons, 1980*). Although widely distributed from Portugal to China until the middle of the last century, currently there are two disjunct sub-ranges: a western one encompassing Iberia, France and Sardinia and an eastern one ranging from southwestern Russia to north-western China (*Morales & Bretagnolle, 2022*). It is classified as "Endangered" in Spain (*SEO/Birdlife, 2021*), "Vulnerable" in Europe (*Bird Life International, 2015*; *Burfield et al., 2023*) and as "Near Threatened" globally in

the IUCN World Red List (*Bird Life International, 2023*). The Iberian Peninsula is the core of the western subrange, whose populations are experiencing a dramatic decline in recent years (ca. 50% from 2005 to 2016, *García de la Morena et al., 2018*; *Morales & Bretagnolle, 2022*). Breeding little bustards depend on heterogeneous cereal farmland with fallow fields (*Morales, García & Arroyo, 2005*). As they have an exploded-lek mating system (*Jiguet, Arroyo & Bretagnolle, 2000*), they tend to be spatially clumped, which largely explains their patterns of breeding abundance and distribution at different scales (*Morales et al., 2014*; *Estrada et al., 2016*; *Arroyo et al., 2022*). Therefore, not taking conspecific attraction into consideration may mask some ecological relationships relevant for conservation (*Estrada & Arroyo, 2012*).

## Study area

The study was carried out in the central sector of the Duero basin in the region of Castilla y León (NW Spain, Fig. 1). Climate is continental-Mediterranean with marked oscillations during the year: hot and dry summers, cold winters and rainfall concentrated in spring and autumn (mean temperature 11.7 °C and mean precipitation 461 mm throughout the year influenced by orography; *Agencia Estatal de Meteorología, AEMET*). Although the abundance of little bustards in Castilla y León has declined significantly in recent times, it still accounts for 5% of the Spanish population (*García de la Morena et al., 2018*).

The limits of the study area were defined in relation to data availability, by adjusting a rectangle encompassing a total of 438 census points sampled in 2016 (12,574.198 km$^2$; see below and Fig. 1). Overall, this area is mainly devoted to agriculture and thus dominated by farmland, although natural habitats are also present as described below. The area includes 15 SPAs (2,395.743 km$^2$): according to the Natura2000 official forms published in 2005, 13 have steppe habitat suitable for little bustards, while the remaining two encompass mainly mountain and riparian habitats (Table A1). Two of the 15 SPAs are also classified as nature reserve: Lagunas de Villafáfila and Riberas de Castronuño, but only the former has potential habitat for little bustards, while the latter includes mainly riparian habitats. SPAs have Management Plans (*Junta de Castilla y León, 2022*) detailing guidelines and recommendations to reach their conservation aims but no specific restrictions in terms of farming practices, although farmers may sign voluntary agreements under Agri-Environmental Schemes (*Castilla y León regional Government, 2015*). The measures promoted for little bustard conservation (as well as other steppe bird populations linked to long-term fallows, grasslands and shrublands) are the promotion of crop rotation between cereals, legumes and fallows, the reduction of agrochemicals and coated seeds, the maintenance of areas with natural vegetation (such as shrubs, field margins, wastelands and grasslands), the delay of mowing-harvesting until mid-July, and the reduction of mortality due to non-natural causes (*e.g.*, limitation of night ploughing or harvesting). Plans also encourage monitoring programmes to assess the species' response to conservation measures (*Junta de Castilla y León, 2022*). Unfortunately, although all these areas have a common purpose, agricultural management is not homogeneous across SPAs: while most of them are experiencing strong agricultural intensification, some undergo a process of land abandonment with much less cover of cereal crops (Table A1).

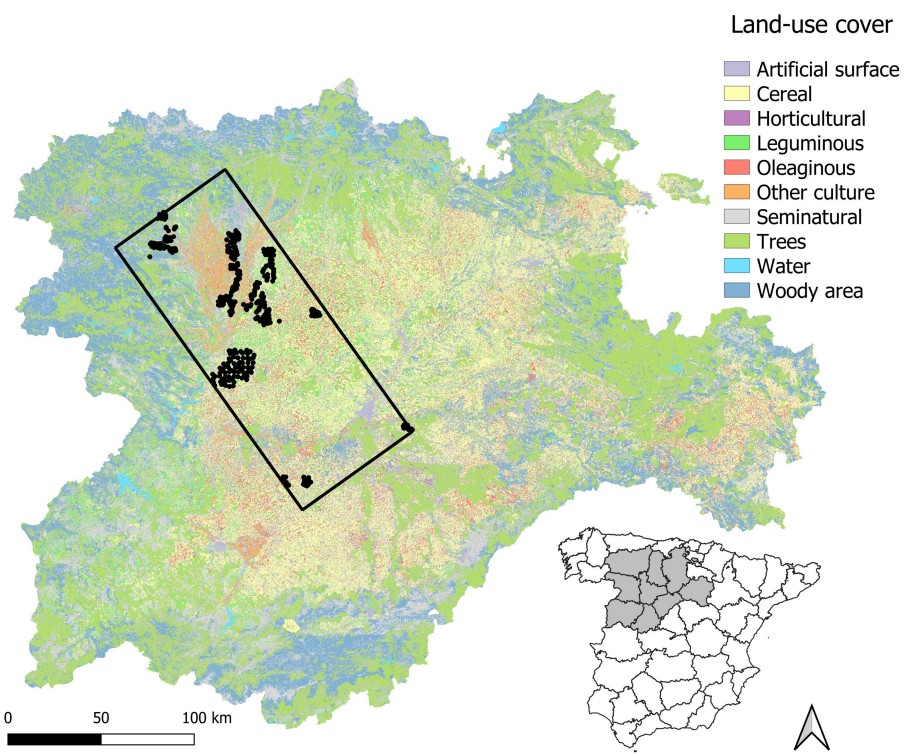

**Figure 1 Map of the study area.** Census points are presented as dots and the rectangle represent the limits of the study area (see Methods).

The Nature Reserve of Lagunas de Villafáfila (hereafter Villafáfila) occupies 325.49 km$^2$ of flat or gently undulated cereal farmland with a few seasonal semi-endorreic lagoons at an average altitude of 700 metres above sea level. Although most terrain is devoted to cereal cultivation, nearly 10% of the reserve is cultivated with dry alfalfa crops for haying and sheep grazing (*Rodríguez Alonso & Palacios Alberti, 2006*). This area is actively managed for conservation according to its Natural Resources Management Plan (*Castilla y León regional Government, 2005*) and any measure implying an intensification of farmland practices must be studied and approved by the park administration. Irrigation or afforestation are forbidden. Management carried out in the nature reserve has an important focus on cereal steppe habitats and their biodiversity (*Rodríguez & Palacios, 2021*). For example, most cereal fields are covered by Agri-Environmental Schemes (hereafter AES), and thus not harvested until mid-July to allow successful breeding of birds. Indeed, more than 60% of the Reserve's extent was under AES in the 2000's (*Rossel & Viladomiu, 2005*). The Nature Reserve of Riberas de Castronuño (84.21 km$^2$, 697.34 m above sea level) is mainly occupied by riparian habitats, although the 35% of its extension is devoted to extensive agriculture. As a nature reserve and part of Natura2000 network, its management seeks to conserve its natural values (*Castilla y León regional Government, 2000*); however, as farmland is under-represented inside the reserve, most conservation measures are focused on preserving riparian biodiversity.

Finally, in unprotected productive regions management is relatively intensive with regular use of pesticides and fertilizers (*Albiac et al., 2017*), cereal harvesting may occur from early July onwards (*Rodríguez-Teijeiro et al., 2009*), fallow land is ploughed several times per year, and the overall area left as fallow is increasingly smaller. In these areas, the proportion of land under AES is much smaller than within SPAs.

## Little bustard data

We used little bustard data at different spatial and temporal resolution obtained from two sources providing comparable data: the national little bustard census carried out in 2016 (*García de la Morena et al., 2018*) and censuses carried out in Villafáfila annually from 2011 to 2020. In both cases, abundance surveys were done from point counts during the breeding season following the same methodology, described in *García de la Morena et al. (2006)* and *Cabodevilla et al. (2020)*. In the annual censuses in Villafáfila, 72 points distributed throughout the nature reserve (avoiding large roads, villages or woodlands) were surveyed each year by regional wildlife officers (the points were the same every year). In the national census of 2016, 20 census points were distributed in 5 × 5 km squares avoiding, as done in Villafáfila, areas where the species is unlikely to be found. A total of 366 points were surveyed in central Castilla y León as part of the national census protocol, none of which was within Villafáfila (as it was already surveyed by the wildlife managers). The number of observers participating in both Villafáfila annual censuses and the 2016 national census in central Castilla y León was high, and all of them were experienced ornithologists. Each point in both data sources was separated at least 600 m from the nearest one and all the individuals visually and acoustically detected in a 250 m radius during a 5-minute period were recorded. Although females were also recorded when detected, most observations corresponded to males due to their much higher detectability. For instance, the annual censuses at Villafáfila (2011-2020) yielded a total of 224 males and 17 females. Because of that, we only used males in analyses. In any case, the number of females recorded was correlated with the number of male observations (see: Table A4 and Fig. A12), and thus "male presence" and "male abundance" are indicators of "species presence" and "species abundance", respectively.

## Land cover information

Land cover information was obtained from the Instituto Tecnológico Agrario de Castilla y León (*ITACYL, 2021*), which publishes annual raster maps (since 2011) with high resolution (20 × 20 m, 10 × 10 since 2017) based on satellite imagery. The methodology used for the identification of different land covers is based on an automatic learning algorithm that uses additional information such as LIDAR data, terrain elevation and slope, or field data. ITACYL considers a high number (more than 25) of land cover categories, many of which are rare and/or not present every year in our study area (*e.g.*, water bodies, trees, other cultures or horticultural areas represent less than 1% on average), although the accuracy of some of those categories has increased with time (so they are considered more often in later than earlier years). Therefore, we finally considered 6 land-use variables for analyses, grouping several land-cover categories according to their functional meaning for little

bustards (*i.e.,* in terms of habitat selection, see review in *Traba et al., 2022*; Table A2). Irrigated crops were present only in some years and in less than 1% of 1 × 1 km squares of the study area during the period analysed. Therefore, we decided not to consider them separately and grouped them with their equivalent rain-fed crop categories (Table A2).

We calculated the proportion of each of the six land cover categories within a 250 m radius buffer around each census point, as well as the Shannon index for land cover diversity as an indicator of landscape heterogeneity. A 250 m buffer has been used in previous little bustard studies and represents the radius where detectability of the species is highest (*e.g.*, *García de la Morena et al., 2018*; *Faria & Morales, 2018*).

## Statistical analyses
### Favourability modelling

To estimate habitat or land-use favourability, we computed a generalized linear model (GLM), fitted to a binomial error distribution, with little bustard male presence as response variable using the data from 2016 (438 census points), which combined the data from the second national census inside the study area with the 72 obtained in that year's census in Villafáfila. Little bustard males were only present in 30 of them (15 within Villafáfila).

As explanatory variables, we included the six land-use categories plus the Shannon index for land-use diversity. We initially considered including a spatial factor (resulting from a polynomial trend surface analysis) in the initial model selection process, following the procedure in *Estrada et al. (2016)*. However, the spatial factor had a very strong impact on probability of occurrence (Table A3, model 1), while we were specifically interested in calculating land-use favourability without constraints imposed by the species' current distribution (which may be influenced by historical rather than ecological factors; Table A3, model 2). Therefore, we finally decided not to include the spatial factor neither the conspecific attraction in the model, which allows identifying favourable areas according to land use, that could be potentially colonized by little bustards (*Acevedo & Real, 2012*; *Chamorro et al., 2021*).

We assessed multicollinearity of the explanatory variables using the variance inflation factor (VIF) between the land cover percentages inside the 250 m radius and the Shannon index resulting from them (Table A2). Since all VIF values of the variables analysed were lower than 5 (Table A2), multicollinearity was not considered an issue and all of them were included in a stepwise model selection procedure (after being standardised as (value-mean)/SD) based on AIC using the stepAIC function of the MASS R package (*Venables & Ripley, 2002*). In this procedure all possible combinations of explanatory variables are analysed, and the best model is selected based on its AIC value. In each step, a model is revised starting with a model that includes all the explanatory variables: its AIC value is calculated and compared with values from the models obtained by eliminating each variable already included and adding the ones not included. The combination with the lowest AIC value was considered the best model (see *Real, Barbosa & Vargas, 2006*; *Acevedo & Real, 2012*; *Estrada et al., 2016* for the same approach in other favourability modelling studies).

Model performance was measured by means of the area under the Receiving Operator Curve (AUC), whose values vary from 0 in completely inaccurate models to 1 in perfectly accurate ones (*Manel, Ceri Williams & Ormerod, 2001*; *Mandrekar, 2010*; *Gonçalves, Cortez & Moro, 2020*).

The favourability function is preferred to simple probability of occurrence because it accounts for differences in prevalence (*Acevedo & Real, 2012*). Favourability values vary between 0 and 1; where values closer to 1 indicate a higher probability of occurrence than expected from chance given the prevalence, whereas values closer to 0 indicate probability of occurrence lower than expected given the prevalence(*Real, Barbosa & Vargas, 2006*; *Acevedo & Real, 2012*). Thus, favourability values are directly comparable across areas or years even if prevalence varies. Favourability scores were obtained from the logistic regression probabilities computed for 2016 as follows:

$$F = \frac{\frac{P}{(1-P)}}{\frac{n_1}{n_0} + \frac{P}{(1-P)}} = 1 - \frac{1}{1 + exp^y}$$

$P$ is the probability calculated by the logistic GLM, $n_1$ is the number of census points where the species is present and $n_0$ is the number of census points where it is absent. Using the getModEqn function (modEVA R package; *Barbosa et al., 2013*) the favourability function was estimated for 2016, and then extrapolated annually from 2011 to 2020 with $1 \times 1$ km resolution based on each year's habitat composition. To validate the biological performance of our land-use favourability function, favourability values computed for 2019 were compared with accumulated locations of GPS-tagged little bustards in the same year (Fig. A1).

To test for differences in favourability values and trends across areas with different level of protection, we computed a Gaussian generalized linear model (GLM) with land-use favourability in each $1 \times 1$ km squares of the whole study area (Figs. A2–A11) as response variable, and year (continuous standardized variable), level of protection and the interaction between both as independent variables. For this model, we discarded those $1 \times 1$ km squares with favourability lower than 0.2, as this is the favourability threshold usually considered to identify areas unsuitable for the target species (*Muñoz et al., 2005*; *Coelho et al., 2018*).

### Little bustard abundance and trend analyses

Using data from the 2016 census ($n = 438$ census points), we compared little bustard relative abundance among the three levels of protection (non-protected areas, SPAs and nature reserves) using a GLM with number of males per census point as response variable (fitted to a Poisson error distribution with a log-link function) and protection level as explanatory variable.

To estimate the influence that favourability has over little bustard population trends we used the census data carried out at Villafáfila from 2011 to 2020 (72 census points each year). A generalized linear mixed model (GLMM) with the number of males in each census point as response variable (fitted to a Poisson error distribution with a log-link function) was implemented. As explanatory variables we considered year (as a continuous standardized variable), the favourability value in the point buffer in that year, and the

number of neighbouring males to account for conspecific attraction. The "number of neighbouring males" represents the number of males counted in other census points within a 1.7 km buffer (mean distance between census points plus the standard error) around each census point. Under the appropriate visibility conditions, little bustard males were detectable up to 1 km away during censuses (*Wolff et al., 2001*). However, presumably little bustards can detect their conspecifics from farther distances, so by using this radius we ensure that conspecific attraction is captured by this variable (*Jiguet, Arroyo & Bretagnolle, 2000*; *Morales et al., 2014*). Census point identity was included as a random intercept.

In all models, we checked normality of residuals using q-q plots. We present ANOVA type III results for the significance of each variable. Mean values and confidence intervals are presented in plots. All analyses were carried out with R software version 4.0.1.

## RESULTS

### Land-use favourability

The final model obtained included three land-use variables: seminatural areas, cereal crops and legume crops. The parameter estimates from this model indicate that male presence increased with the availability of all these three land uses (Table 1). However, cereal crops alone never led to highest favourability values, whereas areas dominated by seminatural vegetation can reach highest favourability values (Fig. 2). To test for spatial autocorrelation, we computed Moran's test on the calculated quantile residuals of this model, demonstrating that autocorrelation was not an issue ($p = 0.830$). The model AUC value was 0.731, which indicates good model performance (*Mandrekar, 2010*; *Gonçalves, Cortez & Moro, 2020*), despite the small variance explained (Table 1).

The favourability function obtained was:

$$F = 1 - \left( \frac{1}{1 + exp^{-0.437 + 1.012 * seminatural + 1.287 * cereal + 0.741 * legume}} \right)$$

Using this function, we calculated the land-use favourability values each year for the whole study area for each $1 \times 1$ km square (Figs. A2–A11). The comparison of favourability values and locations of birds tagged with GPS tags in 2019 showed that little bustards use grid cells with high favourability values, which validates the biological relevance of our final model (Fig. A1).

A GLM analysis showed that overall land-use favourability values in non-protected areas were lower than in protected areas, but that values in the nature reserve were significantly greater than in other SPAs (Fig. 3). Additionally, favourability values tended to increase from 2011 to 2020 for the three levels of protection, although the slope of increase was less pronounced in nature reserves (Table 2 and Fig. 3). Overall, protection level was more important explaining variation in favourability than temporal trends, and the low $R^2$ of the model indicates that other variables not included in our analyses are also important in explaining variation in favourability. The Moran's I test for the calculated mean of quantile residuals for each cell included in the model showed non-significant spatial autocorrelation ($p = 0.494$).

**Table 1** **Results of the final model explaining the occurrence of male little bustards in sampling points in relation to each land use included in the model (seminatural areas, cereals, and legumes).** Explanatory variables were the percentage of each land use (standardised prior to the analyses) in a 250 m radius around the census point. The area under the curve (AUC) for this model was 0.73, all degrees of freedom were 1, and the global R2 was 0.11. $\chi^2$ represents the significance of likelihood ratio chi-squared statistics for each value. The sample size was 438 census points spread over the whole study area (Fig. 1).

| Response variable | Explanatory variable | Estimate | $\chi 2$ | P | $R^2$ |
|---|---|---|---|---|---|
| | Seminatural area | 1 | 12.8 | <0.001 | 0.08 |
| Presence of males | Cereal | 1.3 | 16.6 | <0.001 | 0.10 |
| | Legume | 0.7 | 6.9 | 0.008 | 0.05 |

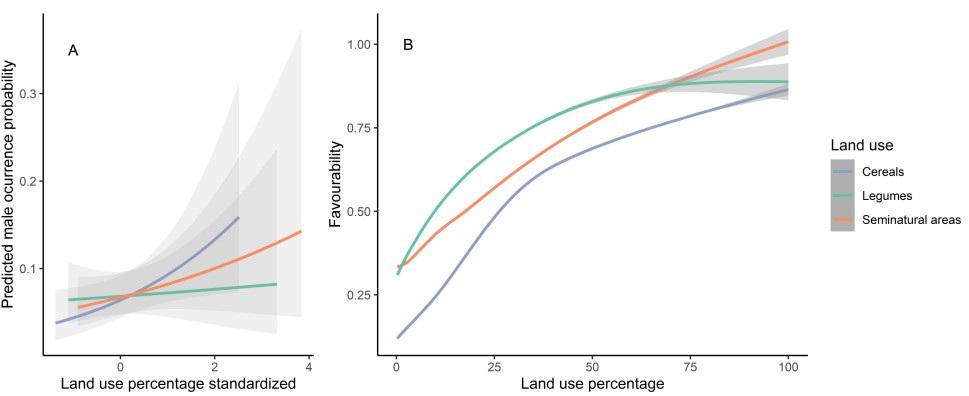

**Figure 2** **Predicted male occurrence probabilities according to the land uses used to determine habitat favourability (A) and the relationship between the favourability values obtained and each land use (B).** Land-use percentages were calculated inside the 250 m radius around the 438 census points spread over the study area. Mean and 95% confidence intervals are shown. Note that although the variables were standardized prior to the analyses, the right-hand panel shows original percentages for ease of interpretation.

**Figure 3** **Plot of values predicted by the final model analysing favourability trends across the study area between the three levels of protection considered.** Note that "year" was standardized prior to the analyses, although the figure presents year labels for clarity. In the plot, mean values and 95% confidence intervals are presented.

**Table 2** **Results of the model analysing the effect of the year, level of protection and their interaction on the land-use favourability computed for the whole study area.** The study period was from 2010 to 2019, and the variable year was standardized prior to the analyses. All $P$ values for explanatory variables were smaller than 0.001, and the global $R^2$ was 0.06. Sample size was 95,688 $1 \times 1$ km cells.

| Response variable | Explanatory variable | Sum Sq | Degree of freedom | F | $R^2$ |
|---|---|---|---|---|---|
| | Year | 9 | 1 | 218.3 | $4.66e^{-15}$ |
| Favourability | Protection | 201 | 2 | 2,447.6 | 0.05 |
| | Year*Protection | 4.2 | 2 | 51.1 | $1.07e^{-3}$ |

## Little bustard abundance and population trends

Male relative abundance differed among the three protection figures ($\chi 2 = 16.80$, $df = 2$, $p < 0.001$, Fig. 4), being much greater in Villafáfila (0.292, sd = 0.638 males/no of census points) than either in non-protected areas (0.135, sd = 0.659 males/no of census points) or SPAs (0.063, sd = 0.491 males/no of census points). Autocorrelation was not an issue in this model (Moran's I test, $p = 0.164$). Male abundance in Villafáfila during the study period decreased by nearly 50% (Fig. 4). Variables explaining male abundance at each point included the number of neighbouring males and year (Table 3 and Fig. 5). Male abundance in each census point was positively related to the number of neighbouring males, increasing markedly when there were more than 4 neighbouring males, and, overall, declined during the study period (Table 3 and Fig. 5).

## DISCUSSION

Little bustard occurrence during the breeding season increases with cover of cereal crops mixed with legumes and seminatural areas. As expected, land-use favourability for the little bustard was higher in protected than in non-protected areas, which suggests that the geographic configuration of those protected areas is adequate or that protected areas indeed maintain the availability of adequate land uses. However, land-use favourability values (and relative little bustard abundance) were much higher for the nature reserve level than in other farmland SPAs of the region, which highlights the importance for the species of the conservation-oriented land-use management carried out in areas under this level of protection (see Figs. 3 and 4). Regulations regarding agricultural activities and land management are most restrictive in nature reserves, especially in Villafáfila (see "Study area"). Management plans of the Castilla y León Natura2000 network do seek the preservation of little bustard (and other steppe birds) populations, but their degree of implementation depends on the will of farmers to enrol in Agri-Environmental Schemes. Villafáfila also has a Natural Resource Management Plan that specifically forbids land management practices detrimental for steppe birds, like irrigation or afforestation (*Castilla y León regional Government, 2005*). Given that we only had one nature reserve devoted to steppe bird conservation in our study system, it is not possible to firmly conclude that the higher abundance of little bustards found in Villafáfila is related only to the protection regime and its associated higher land-use favourability, excluding other factors (*e.g.*,

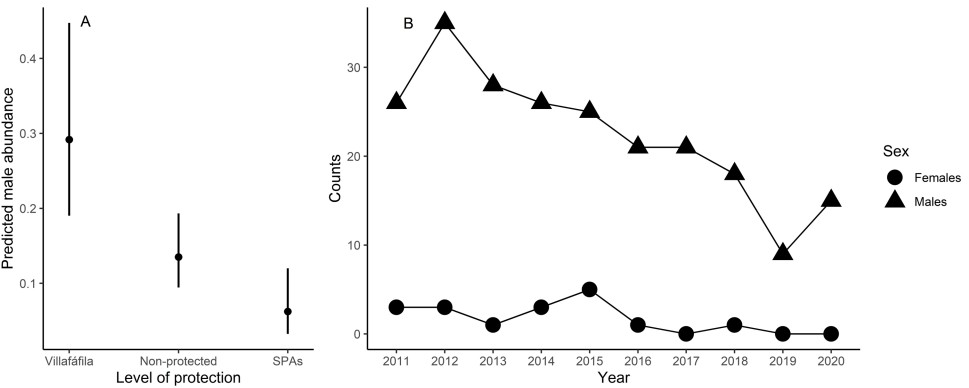

**Figure 4  Results from the model analysing differences in little bustard male abundance between protected area categories in the study area (A) and little bustard trends shown by censuses in Villafáfila (B).**

**Table 3  Results of the model analysing the relationship between the number of little bustard males and favourability, neighbouring males, and year in Villafáfila.** The explanatory variables are the favourability value in the census point, the amount of neighbouring males measured as the number of males in 1.7 km radius, and the year when the census was carried out (standardized prior to the analyses). The $R^2$ for the model was 0.3 and the dispersion value was 0.8. $\chi^2$ represents the significance of likelihood ratio chisquared statistics for each value. Sample size was 72 census points per year.

| Response variable | Explanatory variable | $\chi^2$ | P |
|---|---|---|---|
| | Favourability | 0.2 | 0.683 |
| Number of males | Neighbouring males | 34.8 | <0.001 |
| | Year | 7.2 | 0.007 |

historical events, philopatry), but our results strongly suggest that the much greater land-use favourability found in Villafáfila is likely due to its specifically conservation-oriented management, which includes the promotion of certain habitats favourable to steppe birds, such as rain-fed legume crops (mean percentage of legume crops per $1 \times 1$ km square from 2011 to 2020: Villafáfila = 19%, SPAs = 13% and non-protected = 9%), and limitations to the expansion of other land uses known to be detrimental for them such as natural and cultivated tree cover (mean percentage per $1 \times 1$ km cell from 2011 to 2020: Villafáfila =2%, SPAs =9% and Non-protected =17%).

The rank of average favourability values (Nature Reserve >SPAs >unprotected areas; Fig. 3) found in this study is similar to those found in other studies (*McKenna et al., 2014*; *Martínez-Fernández, Ruiz-Benito & Zavala, 2015*), and supports that the Natura2000 network may not be entirely efficient to attain the conservation goals for many species (*Palacín & Alonso, 2018*; *Gameiro et al., 2020*). Although the Natura2000 network has contributed to preserve European biodiversity (including steppe habitats), its effectiveness largely relies on the area covered by (voluntary) Agri-Environmental Schemes. Therefore, greater funding and legal support is required to avoid the so-called "paper park" effect in Natura2000 areas. The best protection against land-use changes is apparently found in

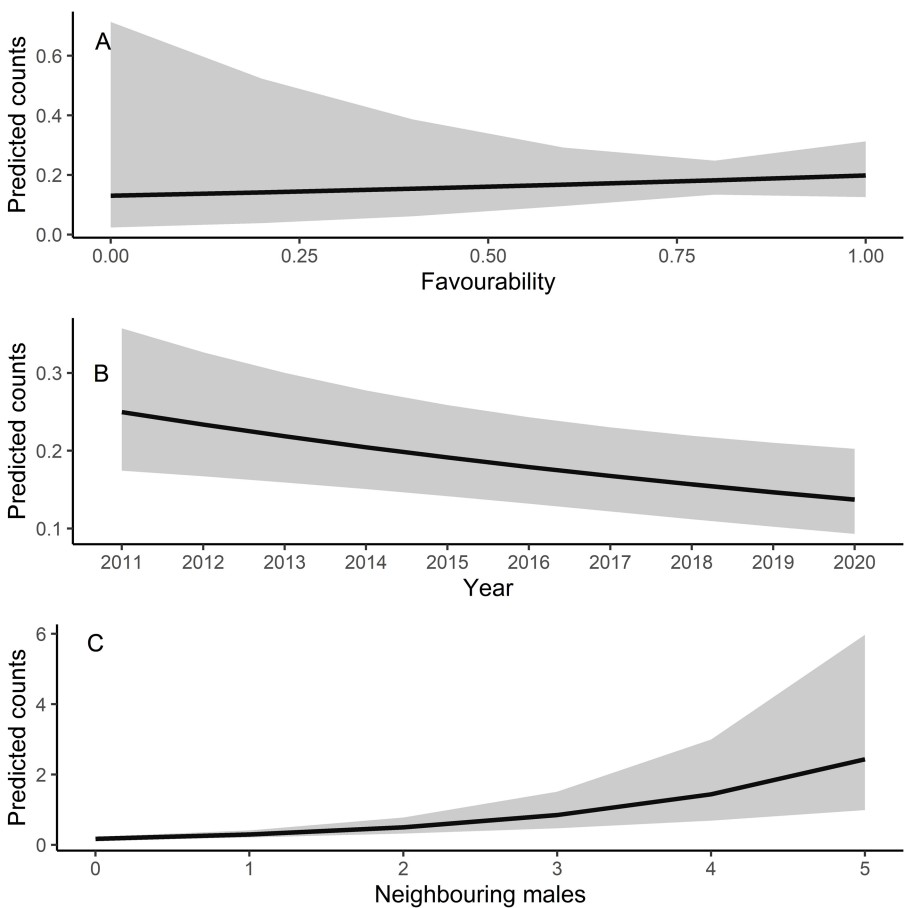

**Figure 5 Predicted counts for little bustard males obtained from the model analysing census data from Villafáfila and its relationship with each explanatory variable.** (A) Shows the relationship between predicted male counts with favourability, (B) with year and (C) with the neighbouring males (*i.e.,* the number of nearby males within a 1.7 km radius around each observation). Mean values and 95% confidence intervals are shown.

nature reserves, probably because of their legal stringency and management regulations (*Rodríguez-Rodríguez & Martínez-Vega, 2018*). Farmland in nature reserves presents a greater proportion of permanent natural habitat (*i.e.,* pastures, meadows) than farmland in Natura2000 SPAs (*Martínez-Fernández, Ruiz-Benito & Zavala, 2015*). However, to avoid isolation in a matrix of unprotected landscape and thereby suffer deleterious edge effects, nature reserves should be surrounded by buffer zones in which a similar management regime is implemented (*Martínez-Fernández, Ruiz-Benito & Zavala, 2015*; *Rodríguez-Rodríguez & Martínez-Vega, 2018*), especially those of small size, since they are more prone to edge effects from the management of surrounding areas. For example, Villafáfila is surrounded by intensive farmland which may reduce the efficiency of the steppe-bird conservation-oriented management of the reserve (*Rodríguez & Palacios, 2021*). Isolated reserves may fail protecting endangered species, particularly those exhibiting far-reaching seasonal movements such as the little bustard (*García de la Morena et al., 2015*). This
is illustrated by records of tagged birds, which tend to move between areas with high favourability (Fig. A1). If highly favourable areas are few and far apart, then little bustards may not find them, resulting in negative population trends as a consequence of Allee effects or increased mortality during movement (*Morales, Bretagnolle & Arroyo, 2005*; *Marcelino et al., 2018*). Thus, the management guidelines mentioned above and implemented in nature reserves like Villafáfila should be encouraged (*e.g.*, with more voluntary contracts) in SPAs to overall increase land-use favourability and connectivity between areas with high quality little bustard habitat.

Land-use favourability slightly increased from 2011 to 2020 in areas with the three levels of protection (Table 2 and Fig. 3). This result was unexpected because Iberian little bustard populations strongly declined during the same interval (*Morales & Bretagnolle, 2022*). However, the increase in favourability values concur with the agricultural changes observed in Spain: in the case of our study area, the percentage of cereal crops has increased during this period in all the levels of protection analysed, which may partly explain the observed favourability trends. Moreover, areas with low crop yields have been abandoned (*Oñate, 2005*), which usually leads to an increase of natural vegetation habitat within the farmland matrix (long-term fallows, grassland, and wastelands). Although farmland abandonment is mainly taking place in areas with moderate to steep slopes, the flat farmland that dominates our study area has a certain cover of natural grasslands and wastelands in lower yielding sectors, which may have contributed to the favourability increase observed, given that these land uses weight positively on little bustard land-use favourability (Fig. 2). However, it is important to emphasize that global habitat quality is likely to be strongly affected by the agricultural practices implemented on-field, beyond the cover of each specific land use type, something that we could not take into account (due to the unavailability of field-level information on management practices) but which would be important to address in future studies. For example, a potential increase in the use of pesticides in cereal crops leading to lower food availability and thus habitat quality for birds cannot be reflected in our analyses. Because of this, our results on land-use favourability trends may be overoptimistic in terms of global habitat quality.

In spite of the greater favourability of nature reserves, and the slight positive trend in favourability throughout the study period also observed in this area, the species also markedly declined there (Fig. 4). Therefore, although the large-scale decline of little bustards and other steppe birds has been found to be associated with the loss of key habitats in the agricultural landscape (*i.e.,* fallow land, see *Traba & Morales, 2019*), our result suggests that the decline in the study region may not right now be driven primarily by landscape changes, and underlines the need to develop finer-scale models of habitat quality accounting for agricultural practices at field level to better understand causes of mortality and breeding failure (see *Bretagnolle, Denonfoux & Villers, 2018*; *Cuscó et al., 2020*). However, it is important to emphasize that the population declined even more outside Villafáfila (*García de la Morena et al., 2018*), which, again, indicates that favourability is a reasonable measure of relative habitat quality.

Little bustards are affected by different threats in addition to agriculture, such as linear infrastructures causing habitat fragmentation and leading to isolated patches that are

difficult to reach by dispersing individuals (*García de la Morena et al., 2007*), or power lines, which generate mortality hotspots (*Silva et al., 2014*). Little bustards may also be constrained by interspecific competition, for example with great bustards (*Otis tarda*), which have similar habitat requirements (*Tarjuelo et al., 2017b*; *Tarjuelo et al., 2017a*). These factors may be affecting little bustard population dynamics at a larger scale, which may be reflected in the negative trends observed in Villafáfila despite its higher land-use favourability. All these factors need to be addressed from a conservation perspective.

Due to their lek breeding system (*Jiguet, Arroyo & Bretagnolle, 2000*), little bustards tend to have clumped spatial distributions. In fact, according to our results, presence of other males is even more important than habitat *per se* in explaining spatial variations in abundance within Villafáfila, something already highlighted in previous studies (*e.g.*, *Morales et al., 2014*). This conspecific attraction also results in females clustering around males (see Table A4, Fig. A12 and *Tarjuelo et al., 2013*; *Morales et al., 2014*), which supports the idea that results obtained from male abundance at landscape scale can be extrapolated to both sexes (*Devoucoux, Besnard & Bretagnolle, 2018*). However, because of this tendency to cluster, little bustards may be extremely sensitive to local extinctions, being absent from certain areas of good quality habitat, as suggested by the lack of a significant relationship of male abundance with land-use favourability obtained from the censuses at Villafáfila (Table 3 and Fig. 5). As they tend to aggregate, it may be difficult for them to colonise new areas even if they are favourable. The land-use favourability of these areas can only be identified by analysing the little bustard—habitat associations independently of behavioural and social factors like conspecific attraction (Table A3). Local population trends may thus be affected by processes occurring at metapopulation scale, since individuals spend most of the annual cycle out of breeding areas (*Morales et al., 2022*). Features such as habitat quality or mortality in the non-breeding quarters may thus have an impact on breeding populations. It is therefore crucial to develop conservation strategies that protect summering and wintering quarters and distribution ranges as a whole in a more geographically integrated manner.

## CONCLUSIONS

Here we demonstrate that little bustard habitat quality resulting from the management of SPAs is poorer than in more conservation-stringent areas such as the nature reserve of Villafáfila. However, little bustard populations have declined even in those protected areas (Fig. 4; *e.g.*, *García de la Morena et al., 2018*; *Morales & Bretagnolle, 2022*). Furthermore, the population has declined despite the maintenance or slight increase in land-use favourability estimated for the three levels of protection considered. Although the latter may be partly related with the fact that models did not account for field-scale factors such as farming practices, these results suggest that the little bustard decline could be steepened by some behavioural traits of the species associated to lek mating, such as conspecific attraction or the density dependent space use shown by our models (Table 3 and Fig. 5).

The high land-use favourability values found in nature reserves like Villafáfila (the other nature reserve is not mainly focused on steppe birds) suggest that they are

likely a consequence of their active management focused on steppe bird conservation (*González del Portillo et al., 2021*). This contrasts with SPAs and, particularly, non-protected areas which, overall, cannot be considered favourable for the species (Figs. A2–A10). This highlights the need to increase the level of conservation-oriented landscape management outside protected areas (ideally, at the level attained in nature reserves like Villafáfila), particularly in those sites where the species is still present, to ensure the preservation of the little bustard. Our results (Fig. 2) showed that the highest values of favourability can only be reached if seminatural areas are abundant in the farmland matrix. Therefore, promoting fallow fields and other land-uses with natural herbaceous vegetation cover would increase favourability, but also population productivity if adequately managed, as they are preferred by little bustard females as main nesting habitat (*Morales et al., 2013*).

## ACKNOWLEDGEMENTS

We would like to thank to all the field assistants who carried out the censuses in the Nature Reserve of Lagunas de Villafáfila (in alphabetical order): Ana Martínez Fernández, Cayetano Caldero Prieto, Cristian Osorio Huerga, Eduardo Vega Rábano, Emilio Álvarez Fernández, Hipólito Hernández Martín, Jesús Domínguez García, José Javier Orduña Justo, José Luis Gutiérrez García, José Miguel San Román Fernández, Juan Moran Blanco, Luis Fernando San José Luengo, Luis Pintado García, Ma José Rodríguez Ferrero, Manuel Ángel Fidalgo Centeno, Manuel Hernández Jaspe, Manuel Miñambre Fidalgo, Manuel Segura Herrero, Pablo Santos Redin, Pedro Diez Iglesias, Roberto Gómez Mezquita, Roberto Montero Asensio, Sergio Domínguez Rodríguez, Tomas Yanes García and Vicente Fernández Martínez; and specially to Mariano Rodríguez Alonso and Jesús Palacios Alberti as the coordinators of the project. We also want to acknowledge the contribution of the Second national census as we used its data for this article.

### Funding

This study has been funded through the Memorandum of Understanding between the International Fund for Houbara Conservation (IFHC) and Fundación Patrimonio Natural (FPN) de Castilla y León, as well as through the Chair UAM-CTFC-TotalEnergies Steppe-Forward. David González del Portillo's contract was financed by Comunidad de Madrid through the Youth Employment Initiative programme of the European Social Fund (contract no. PRED_16048). There was no additional external funding received for this study. The funders had no role in study design, data collection and analysis, decision to publish, or preparation of the manuscript.

### Grant Disclosures

The following grant information was disclosed by the authors:
International Fund for Houbara Conservation (IFHC).
Fundación Patrimonio Natural (FPN) de Castilla y León.
European Social Fund: PRED_16048.

## Competing Interests

The authors declare there are no competing interests.

## Author Contributions

- David González del Portillo conceived and designed the experiments, performed the experiments, analyzed the data, prepared figures and/or tables, authored or reviewed drafts of the article, and approved the final draft.
- Manuel B. Morales conceived and designed the experiments, performed the experiments, analyzed the data, prepared figures and/or tables, authored or reviewed drafts of the article, and approved the final draft.
- Beatriz Arroyo conceived and designed the experiments, performed the experiments, analyzed the data, prepared figures and/or tables, authored or reviewed drafts of the article, and approved the final draft.

## Data Availability

The land-use cover datasets analysed in the current study are available at ITACYL: https://mcsncyl.itacyl.es/descarga.

The census datasets available in the Supplemental Information are property of the Nature Reserve of Lagunas de Villafáfila (Junta de Castilla y León) and SEO-BirdLife (Second National Census).

## Supplemental Information

Supplemental information for this article can be found online at http://dx.doi.org/10.7717/peerj.16661#supplemental-information.

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
