# Peer review of "Temporal trends of land-use favourability for the strongly declining little bustard: assessing the role of protected areas"

_PeerJ, doi:10.7717/peerj.16661_

## Round 0.1 · original submission · Major Revisions

The reviewers had somewhat different takes on your manuscript. For unknown reasons, many invites to review this study were refused, so I also read the manuscript and had a variety of suggestions. Essentially, you have three reviews, and I would like you to pay close attention to all three. Mine is in the annotated manuscript.

Both reviewers also had issues concerning the writing style and clarity. Please pay close attention to all the grammar and writing style comments, and try to infer patterns so that you can improve the text in other places. One reviewer called attention to the analyses and I agree - you should explain somewhat more, and perhaps describe more clearly.

My annotated copy addresses many writing style issues. Your style can be much more succinct and clear. Clarity is of the utmost importance because if you're not clear when explaining your results, the reader wonders exactly what you did, and they can become doubtful of the results. I hope you find all of our comments helpful in revising this conservationally and ecologically interesting manuscript.

Reviewer 1 ·

Basic reporting

No commment.

Experimental design

I found it somewhat difficult to follow how the bird censuses were conducted. I suggest including a few more details, which I mentioned elsewhere.

Validity of the findings

No comment.

Additional comments

Dear editor,
I finished my review of the manuscript entitle “Temporal trends of land-use favourability for the strongly declining little bustard: assessing the role of protected areas”, submitted to PeerJ. It is an important contribution to the understanding of the decreasing population of the Little Bustard in Spain, articulating its occurrence with protected areas and landscape composition.
It certainly merits publication, but before that, I suggest a few contributions which I would like the authors to consider.

Line 54: update reference. Suggestions: DÍAZ S ET AL. 2019. Pervasive human-driven decline of life on Earth points to the need for transformative change. Science 366(6471): eaax3100.

Lines 61-65: I am not acquainted with European protected areas. Therefore, for a wider audience, I suggest including a few details about them. I rapidly searched and found several papers dealing with the protected areas system in Europe. Specifically, is there a connected network of state, federal or international protected areas? There is a lot of information about this issue for Spain, but considering the Little Bustard inhabits other countries, I recommend including details about this.

Line 86: Did the authors consider evaluating the occupancy and detection probabilities models by MacKenzie and colleagues? By doing so, one can account for imperfect detection, i.e., when the probability of detecting a species is less than 1.

Line-94-108: I recommend including that the species is near threatened at the global level according to the IUCN.

Lines 109-123: I suggest clearly indicating in the objectives where the species was censused so the reader can immediately know the scale of the study.

Lines 127-143: I found a little misleading and repetitive to write about the species again in Material and methods since there is some specific information about it in the Introduction. Perhaps in the Introduction the authors may include not only the Little Bustard as an example for the declining population trends of birds with increasing agricultural land use.

Line 148: Are there values for rain fall and temperatures so the international readers can follow the specificity of the region’s climate?

Line 198: It is not clear how many people censused the birds. Is there more than one? Considering the detectability of each human individual varies, was the number of observers accounted for in model selection?

Lines 198-207: Were all points censused during all years? How can the authors be sure the population decline indicated in the results is not an artifact of sample bias? I suggest a little more detailed regarding how the points were conducted.

Line 323: If these covariables are associated with the occurrence of the birds, is the species really a bioindicator, as mentioned before?

Discussion – Area is presented in Table A1 but it is not used in model selection. It has an important and recognized effect, which states there is higher density in larger areas. Did the authors consider this idea?

Line 374: Does this mean that little bustards should have been encountered LESS on protected areas, since they are supposed to harbor LESS man-made cereal and legume crops?

Line 410: Is it possible to include in Table A1 the matrix habitats of the study areas? Do cereal and legume crops surround the N. R. of Villafáfila?

Line 419: If individuals cross areas, how sure can we be sure that those counted in points separated by 600 m are not being counted again, i.e., how to avoid overcounting them?

Lines 459-464: I am not sure this paragraph is necessary, as it is not related to the authors findings.

Lines 465-475: I believe another important issue to tackle is the presence of competing species. It should be worth mentioning whether other species may occur sympatrically, thus affecting the little bustard population estimates.

Figure 1: is it possible to include vegetation types in the background, instead of an entire grey background?

·

Basic reporting

This article focuses on the abundance and change of Little Bustard in a Spanish region and relates it to changes in habitat suitability due to agricultural intensification and protected areas with and without specific conservation strategies. Overall, the article is well written and the question is interesting, with explicit hypotheses set out in the introduction. The figures are relevant and the raw data is available. The method section could be made clearer by merging the "Species studied" and "Little Bustard data" sections.

Experimental design

The methods are well described but should be improved to support the results and the discussion (see general comments below).

Validity of the findings

The discussion is well detailed but again it is based on results that are not yet fully supported by the current methodology (see general comments below).

Additional comments

Major issues:

My major concerns relate to the methods. The first analysis of favourability is carried out via a stepwise regression model. This type of approach can be relevant, but it has also significant limitations (cf. https://doi.org/10.11648/j.ajtas.20150405.22). Given that there are only 6 possible explanatory variables (land-use types), a model incorporating all 6 variables should be considered.

The second main point is that two censuses are used in this paper but in the main analysis of the paper, the data from the two censuses are merged without taking into account potential biases. I would recommand adding census ID as a random variable in a GLMM to test the effect of protection level on Little bustard abundance and trend. The results currently show an effect of the N.P. of Villafáfila, but this is also where one of the censuses took place, so I am not yet convinced that it is not a spurious effect of census ID.

The third point concerns the analysis of Little bustard trends and their relationships with changes in favourability, which is currently split into two analyses, one for males and one for females. Why not use a single analysis with sex as a random variable? There is only one additional variable for the female model (male in the same census) which could also make sense for males (due to lek mating and territoriality) or at least be non significant. This would make the analysis clearer and give more statistical power to the model since the number of individual is rather low (284 males and 110 females?).

The fourth important point is about spatial autocorrelation. This is currently only assessed in the first analysis concerning favourability. But this issue should also be taken into account for all the other analyses, given that the distribution of the sites is not random.

The final main point concerns the way in which agricultural intensification is assessed in the paper and the lack of control variables. The effect of agrochemical is acknowledged in the paper but not tested even though it is one of the major cause of resource depletion for the Little bustard. I know that the data are not easily accessible for pesticides but data about fertiliser input are available (in one of the cited reference https://doi.org/10.1016/j.scitotenv.2017.03.110) and often correlate with pesticide use.

Minor issues:

Lines 98-100: Only one reference is provided (Morales and Bretagnolle 2021) but "Various studies" is stated.

Line 101-104: In the definition of agricultural intensification, increasing use of chemical inputs (pesticides and fertilisers) should be added (cf. https://doi.org/10.1016/j.agee.2017.11.028). In particular as there effects are mentioned line 108.

Line 107: Mentioning authors' data is not needed, as it is already supported by previous studies.

Line 119: 10 year is a mid-term, not a long term, period for the species considered (equivalent to the species lifetime cf. https://doi.org/10.1007/978-3-030-84902-3).

Line 152: Is there a buffer around the sites that are on the edge of the rectangle? It could be relevant to avoid edge effects in the models.

Line 161: If I understand well,one SPA and one Nature Reserve are classified as non-protected areas, which is odd. Why not focus this analysis on the suitable habitat (this is the argument to set these two protected areas as non-protected areas) either by focusing only on steppe land-use or on areas where the favourability is above 0.5?

Line 177: "m.a.s.l." could be changed to "meters" or explicitly to "metres above sea level".

Lines 189-192: Theses statements could be made more clear by adding numbers (e.g. for proportion of land under AES in protected versus non-protected areas).

Line 194: This subsection could be merged with the above section "Study species" to make the Methods section clearer. The number of individual per year should be stated somewhere (e.g. in supplementary) and the total number of individual (284 males and 110 females?) should be stated in this section.

Line 215: The effect of land cover should be controlled by environmental variables (e.g. temperature and precipitation) and by the input in fertilisers which is also available, see my comment above.

Line 218: Explain how the differences in spatial resolution before and after 2017 have be taken into account in the analyses.

Lines 255-256: This should be stated earlier in the paragraph.

Lines 256-257: This should be move to the results.

Lines 258-259: "which rendered non-significant results (p = 0.830)" should also be moved to the results.

Lines 262-271: This part could be moved to supplementary materials or remove (redundant with the Moran test for spatial autocorrelation).

Lines 281-282: P, n1 and n0 should be in italic.

Line 286: Why comparing with 2019 specifically?

Lines 288-292: Spatial autocorrelation should also be taken into account in this analysis. Check wording for consistency: "level of protection" here and "protection categories" line 295 and detail these categories as in line 296 "(non-protected areas, SPAs and the N.R. of Villafáfila)".

Line 296: if "non-protected areas" include Montes Aquilanos and Riberas de Castronuño, there could be a bias (see my comment above). All non steppe areas could be removed for this analysis or the authors can focus only on areas with favourability > 0.5.

Line 297: The two census ID should be added as a random variable to be able to account for potential differences in monitoring (e.g. observer skills).

Line 308-309: Why not mean + 1.96SE to have 95% CIs ?

Line 323: Would this result be different if the 6 land-use variables were kept?

Line 332: Detail the calculation to go from the first equation to this one in supplementary and specify what is -0.437 in Table 1 (intercept?).

Lines 334-336: It remains to be shown that this is statistically significant in the census area.

Lines 340-341: In addition to model results, time-series of favourability could be added in supplementary materials.

Line 345: Remove "test".

Lines 347-349: To show this, this is particularly important to account for the difference in census ID in the model, as one census is dedicated to NR of Villafafila.

Line 353: Change "increase" by "seems to increase" or something similar (or even remove) as it is not significant.

---

## Round 0.2 · Major Revisions

While your manuscript has improved considerably, I am returning this manuscript to you for major revisions, they are very important revisions. I appreciate your effort in improving the manuscript, and it is much better now, except for the very disconcerting issues I raise in the annotated manuscript.

Briefly, it looks to me that the tables of results of your analyses are all incorrect. Your R^2 (and r^2) values are wrong, F values appear to be wrong, and the conclusions from the analyses appear to be wrong. Also, your figures seem to have issues as well. In all cases, your legends are incomplete. Remember, a figure or table should be completely described by, and interpretable through, the legends. The reader should NOT have to refer to the text to interpret the tables and figures. Make sure you read PeerJ's requirements for both tables and figures (letter should indicate separate panels of the same figure, and they should be described with that letter in the legend).

In your results, you should also clearly describe your trends, and the results of analyses, and a reader should be able to understand your results and the tests without needing to see the figures and their legends.

Remember, if an r^2 value is something like 1.07e-3, that means it is 0.00107, and that means 0.11% of the variance is explained, and 1% is nothing, and certainly cannot be significant at P < 0.001! If the total model R^2 = 0.058, that means the total model ONLY explains ~6% of the variance, and that is too small to be meaningful (in your table 2), and also is unlikely to be significant at P < 0.001.

In the introduction, I provide a variety of suggestions to improve the flow and ease of reading. Please try to note the patterns and apply corrections to similar situations in the text elsewhere. You still have a bit of a tendency towards being too wordy. I recommend that you go back to the original reviews and read what each said about writing style and analyses, and double-check whether their, and my, comments continue to be applicable.

---

## Round 0.3 · Minor Revisions

While your manuscript is clearly improved, and we appreciate the effort, I still must insist on re-examining your analyses. Please see my comments on your table 2. Allow me to explain here. In your first model of field type (table 1), you only explain a total of 4% of the variance in the model. This implies that 96% of whatever influences male occurrence remains unknown. In other words, the model explains such a small part of the variance as to be TOTALLY IRRELEVANT. I emphasize that strongly because you will need to justify relevance to resolve this issue. Remember, statistical significance can come from extremely large sample sizes, but biological significance can only come from the explanatory power of the model! And, explanatory power is explained by r^2!

In table 2, matters are even worse. The model ONLY explains 6% of the variance in favourability, of which 5% (83% of the total) is explained by Protection status, and essentially NOTHING else explains anything. Again, your models explain so little of the variance, that more than 90% of the variance in favorability is unknown and unexplained. As such, your own results imply that your study explains nothing important. The only conclusion from this is that your measure of land use favorability and so on are essentially meaningless for determining presence of the bustards. Table 2 also suggests a sample size of almost 10 thousand! I can't figure out what that means.

In your figure 2A, your confidence intervals and lines overlap so much that the different land use types essentially are the same. None is statistically different from any other. However, your 2B seems to suggest otherwise and the two do not seem to agree with each other.

Finally, your results in figure 3 do NOT seem to agree with your other results, because the confidence intervals of the lines are way too narrow, implying a much stronger effect than your other analyses would suggest.

I repeat that I think something is wrong with the analyses. And, I cannot understand how you go from models that explain essentially nothing (favourability P = 0.683) to your final model that seems to strongly indicate that there is an association between favourability and bustard presence. I get the impression that your study is really finding that measures of favourability are useless to predict bustard abundance, while land use patterns and protected status are better predictors.

Some of your results refer to apparently supplemental information, and that information was unavailable to me. You mention, on line 318, Figures A1-A11. You also mention Figure A1 and I wonder if you meant 1A?

In any case, I think you still need to clarify these issues. I did read your rebuttal, and yet these issues remain.

---

## Round 0.4 · accepted · Accept

While I am still a bit concerned about the analyses, I think we'll let the readers decide. Your article has interesting points that are important for conservation and management. I did not send this version back to reviewers because you've attended well to both their and my suggestions. Please ensure that your figures and tables conform to PeerJ standards.